# Simultaneous construction of axial and planar chirality by gold/TY-Phos-catalyzed asymmetric hydroarylation

Pei-Chao Zhang [1], Yin-Lin Li [1], Jiafeng He [1], Hai-Hong Wu [1✉], Zhiming Li[2✉] & Junliang Zhang [2,3✉]

The simultaneous construction of two different chiralities via a simple operation poses considerable challenge. Herein a cationic gold-catalyzed asymmetric hydroarylation of ortho-alkynylaryl ferrocenes derivatives is developed, which enable the simultaneous construction of axial and planar chirality. The here identified TY-Phos derived gold complex is responsible for the high yield, good diastereoselectivity (>20:1 dr), high enantioselectivities (up to 99% ee) and mild conditions. The catalyst system also shows potential application in the synthesis of chiral biaryl compounds. The cause of high enantioselectivity of this hydroarylation is investigated with density functional theory caculation.

[1] Shanghai Key Laboratory of Green Chemistry and Chemical Processes, School of Chemistry and Molecular Engineering, East China Normal University, Shanghai, China. [2] Department of Chemistry, Fudan University, Shanghai, China. [3] State Key Laboratory of Organometallic Chemistry, Shanghai Institute of Organic Chemistry,  CAS, Shanghai, China. ✉email: hhwu@chem.ecnu.edu.cn; lizm@fudan.edu.cn; junliangzhang@fudan.edu.cn

As an effective and exceptionally versatile strategy, the development of catalytic asymmetric intramolecular hydroarylation for the construction of axial[1–10] and planar[11–13] chirality or helicenes[14–20] has received much attention[21–32]. Various transition metal catalysis involving Au, Pt, Rh, and Pd, have been reported in this regard. For instance, the groups of Alcarazo[1,14,16,18] and Tanaka[6,15,17] extensively studied the Au-catalyzed and Rh-catalyzed intramolecular alkyne hydroarylation to achieve enantioselective synthesis of axially chiral biaryls and helicenes, respectively (Fig. 1a). In 2013, Uemura et al.[10] developed a Pd-catalyzed asymmetric cycloisomerization of enynes for the construction of axially chiral biaryl. The group of Urbano and Carreño[11] developed an Au-catalyzed cyclization to construct the planar naphthalene-fused ferrocenes in 2016 (Fig. 1a). Later, Shibata et al.[12] reported the similar reaction enabled by a Pt-catalysis. Meanwhile, the capacity of intramolecular hydroarylation is impressively complemented by various other versatile stereocontrolled methodologies. For example, Shibata[2] and co-workers reported a Rh-catalysed enantioselective synthesis of axially chiral PAHs via regioselective cleavage of biphenylenes. Stará and Starý[19,20] developed a Ni-catalysed enantioselective [2 + 2 + 2] cycloisomerisation of aromatic triynes to obtain the helicene derivatives. Toullec[3], Yan[4], Irie[5], and Sparr[7–9] independently demonstrated efficient organocatalytic enantioselective cyclization of aryl-alkynes to construct valuable molecules containing axial or axial and helical stereogenic elements.

In this work, the simultaneous construction of axial and planar chiralities is realized via the gold-catalyzed[33–42] asymmetric intramolecular hydroarylation of readily available o-alkynylferrocene derivatives 1[43] (Fig. 1c). And the simultaneous construction of two different types of chiralities is now flourishing development (Fig. 1b)[44–51].

## Results

**Optimization of the reaction conditions.** However, to the best of our knowledge, simultaneous construction of multiple chiralities by asymmetric gold catalysis has not been reported so far, this hypothesis faced considerable challenges: (1) the universal and efficient asymmetric intramolecular desymmetric cyclization for the construction of planar ferrocene derivatives was not well developed (In the work of refs. [11,12], 10–20 mol% catalytic loading was used), (2) on the other hand, simultaneous construction of axial and planar chiral molecules via asymmetric catalysis has rarely been reported[52,53] and innate reluctance to

undergo a related asymmetric desymmetric aromatization, and (3) the inherent difficulties to asymmetric gold catalysis, which stem from its linear coordination geometry and the outer-sphere nature of Au(I)-catalysis[44–50]. To test our hypothesis, our investigation began with the cyclization of ortho-alkynylaryl ferrocene derivative 1aa. A series of commercially available chiral ligands were investigated (please find more details in the Supplementary Information (SI) Supplementary Fig. S3). Unfortunately, catalysts used by Urbano-Carreño[11], Shibata[12], or Uemura[10] all failed to give the desired cyclization product (Fig. 2, entries 1–3). (R)-DTBM-SEGPhos (L1), the commonly used chiral ligand in asymmetric gold catalysis, delivered the enantiomer of 2aa in 15% NMR yield with 80% ee and 3:1 diastereoselectivity. (S,S)-Ph-BPE (L2) could give the product 2aa in quite low enantioselectivity and conversion. (R)-BINAP (L3) furnished 2aa in 62% NMR yield with 51% ee and 5:1 d.r., and the dominant Brønsted acid (R)-CPA (L4) in the field of atroposelective synthesis of axially chiral molecules, cylohexanediamine-derived (S,S)-DACH-Phenyl Trost Ligand (L5), (R)-BI-DIME (L6), Binol-derived phosphor-amidite (Sa,R,R)-CPPA (L7), chiral oxazoline-phosphine ligand (R)-tBu-PHOX (L8), all have insufficient catalytic activity. We next turned to investigate our developed chiral sulfinamide phosphine (Sadphos), which has shown good performance in asymmetric gold catalysis[41–43] (Sadphos are commercially available (Daicel and Strem), which are also easily prepared in 2–4 steps from chiral tert-butane sulfinamide.) (Fig. 2). The gold complexes derived from Ming-Phos[42,43,54–56], Xu-Phos[57–61], Xiang-Phos[62–65], PC-Phos[41,66–68] and TY-Phos[69] could deliver (−)-2aa in up to 83% yield with 85% ee. We found that enantioselectivity roughly correlates to the electrical properties of the Sadphos, with a more σ-donating ligand providing a higher enantioselectivity and catalytic activity. Notably, Au(TY-Phos)Cl could be easily synthesized in a gram scale via a five-steps in "one-pot" synthesis (for more details, see the SI). Inspired by this promising result, we varied the Ar group of TY-Phos (TY2–TY4) and introduced electron-donating groups to the aryl moiety of TY-Phos, structured as TY5-TY6 with alkyl groups. Gratifyingly, the product 2aa could be obtained in 83% yield with 85% ee and 5:1 d.r. with the use of [Au(TY5)]BAr$^F$ as the catalyst. Moreover, N-Me-TY5 delivered much lower enantioselectivity to 46% ee (Fig. 2, entry 10). Then, the effect of the counterion was investigated (Fig. 2, entries 8, 11–13)[70], among which sodium tetrakis[3,5-bis(trifluoro-methyl) phenyl]borate (BAr$^{F−}$) delivered the best enantioselectivity and reactivity. Subsequently, either lowering the temperature to −10 °C or using other solvents such as DCE and toluene failed to give better result (Fig. 2, entries 14–16).

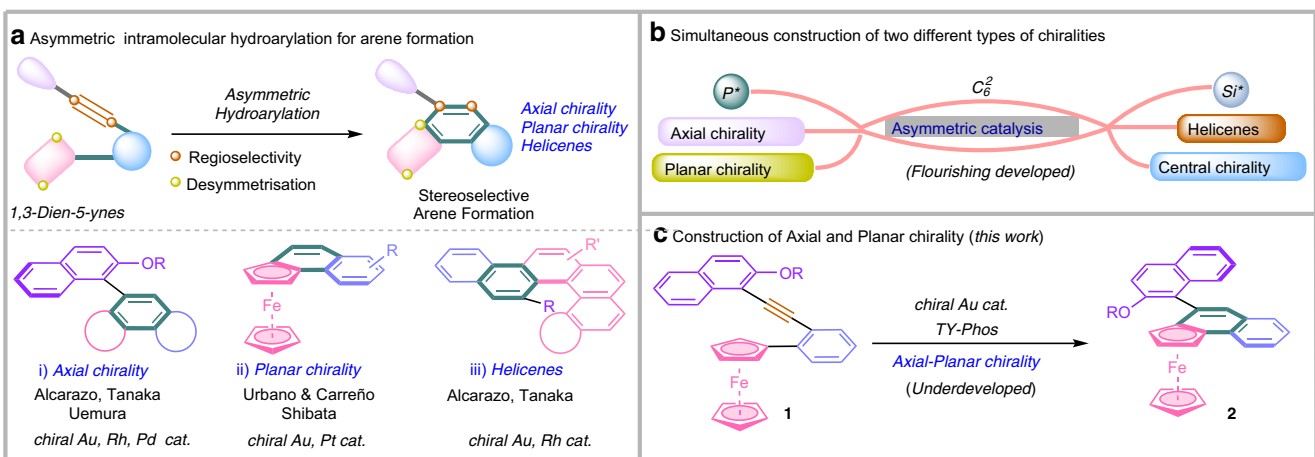

**Fig. 1 Background and project synopsis. a** Asymmetric intramolecular hydroarylation for arene formation. **b** Simultaneous construction of two different types of chiralities. **c** Construction of Axial and Planar chirality (this work).

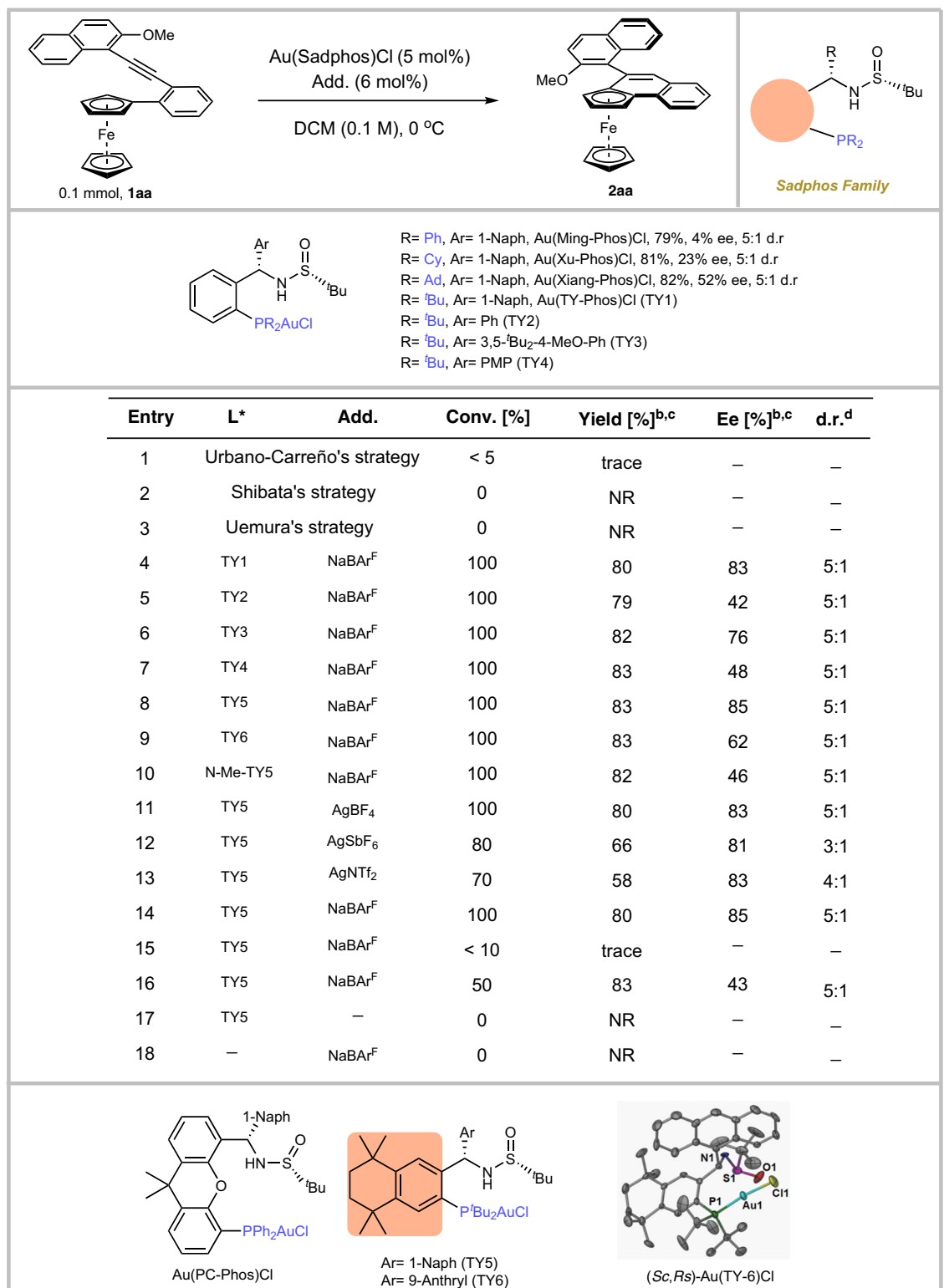

**Fig. 2 Optimization of reaction conditions.** [a]Unless otherwise noted, all reactions were carried out with 0.1 mmol of **1aa** and 10 mol% of catalyst (Au: P: NaBArF (additive) = 1:1:1.2) in 1.0 mL DCM at 0 °C for 24 h; [b]Isolated yield. [c]Determined by chiral HPLC. [d]Determined by NMR. [e]At −10 °C; [f]In 1.0 mL toluene; [g]In 1.0 mL DCE. Shibata's strategy: $PtCl_2(cod)$ (10 mol%), (S,S)-Ph-BPE (10 mol%) and $AgBF_4$ (20 mol%) in DCE at r.t.; Urbano-Carreño's strategy: (R)-DTBM-SEGPhos$(AuCl)_2$ (10 mol%) and $AgSbF_6$ (20 mol%) in toluene at 0 °C; Uemura's strategy: $[Pd(MeCN)_4](BF_4)_2$ (5 mol%) and (R)-Binap (10 mol%) in DCE at 60 °C.

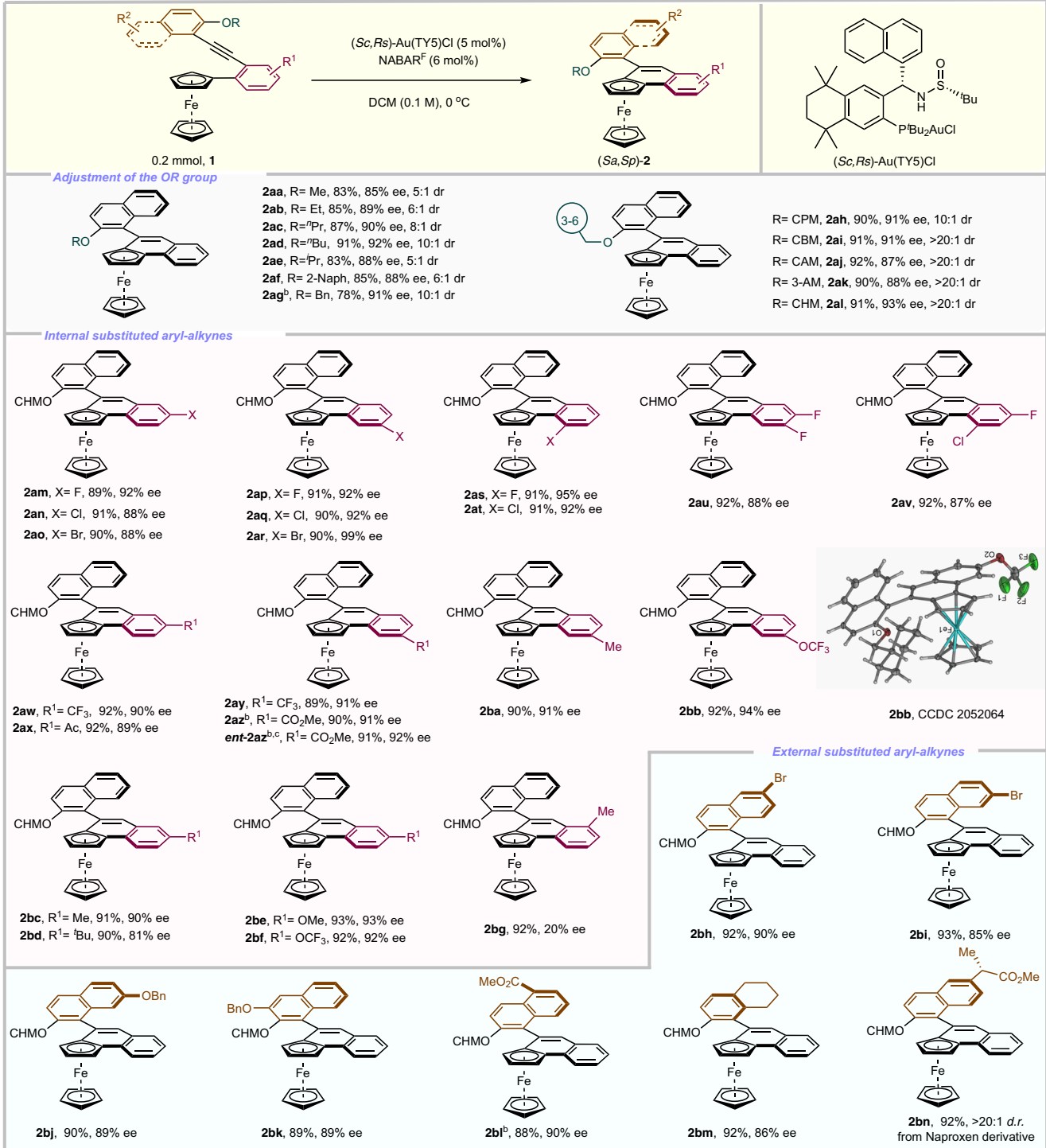

**Fig. 3 Exploration of substrate scope.** [a]Reaction conditions: unless otherwise noted, all reactions were carried out with 0.2 mmol of **1**, Au(TY5)Cl (5 mol %), NaBAr[F] (6 mol%) in 2.0 mL DCM at 0 °C; isolated yields; the ee values were determined by chiral HPLC, >20:1 dr. [b]Au(TY5)Cl (10 mol%), NaBAr[F] (12 mol%). [c](Rc,Ss)-Au(TY5)Cl. CPM Cyclopropylmethyl, CBM Cyclobutylmethyl, CAM Cycloamylmethyl, AM amylmethyl, CHM Cyclohexylmethyl.

Finally, no counterion or only additives which all have not catalytic activity (Fig. 2, entry 18–19).

**Reaction scope study**. Further optimization focused on adjustment of the OR group. Surprisingly, a meaningful increasing enantioselectivity and reactivity and diastereoselectivity occurred when the R was switched to the longer carbon chains (Fig. 3, **2aa**–**2ad**). Bulkier groups (O[i]Pr, O-2-Naph) could furnish the

corresponding products in 88% ee (**2ae**, **2af**). Better enantioselectivity (92% ee) was achieved with a benzyl (Bn) protecting group (**2ag**), however, the diastereoselectivity and reactivity is still far from ideal. The cyclohexylmethyl (CHM) group seems to be crucial to deliver good result (**2ah**–**2al**). A series of internal substituted aryl-alkynes were then prepared and tested to this cyclization process. A good tolerance towards halogens (**1am**–**1av**), electron-withdrawing substituents (**1aw**–**1az**), electron-withdrawing substituents (**1aw**–**1az**), electron-donating

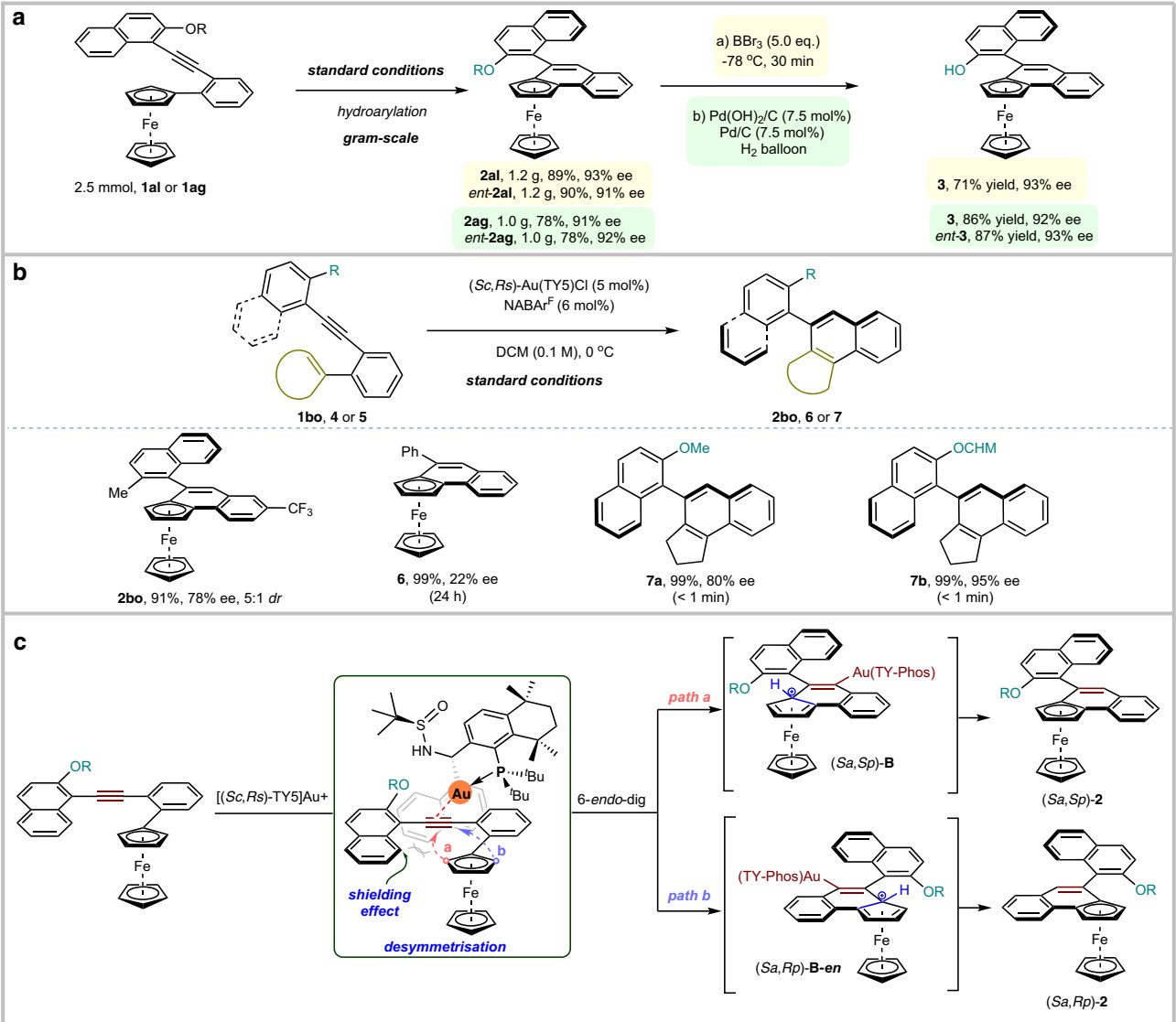

**Fig. 4 Proof-of-principle study. a** The practical utility. **b** mechanism study. **c** proposed asymmetric induction model.

substituents and steric hindrance groups (**1ba–1bf**) on different positions of the phenyl ring ($R^1$) were observed and the corresponding products (**2am–2bf**) were obtained in 88–92% yields with 81–99% ee. The structure and configuration of (*Sa,Sp*)-**2bb** was unambiguously determined via its X-ray analysis. There is an obvious substituent effect at the *ortho*-position of the aryl alkyne, indicating that the steric hindrance at *ortho*-position is unbeneficial to the enantioselectivity (**2bg**). Gratifyingly, the naphthyl and phenyl ring bearing electron-donating and electron-withdrawing groups at different positions ($R^2$), could deliver the desired products (**2bh–2bm**) in excellent yields with moderate to high ee. Then, employing the derivative of pharmaceutical naproxen as the substrate **1bn**, the corresponding product **2bn** could be obtained in high yield with excellent diastereoselectivity.

To demonstrate the practical utility of this protocol for synthesis of ferrocene derivatives bearing axial and planar chirality (Fig. 4a), four reactions were carried out in gram scale under standard conditions. With the use of Au[TY5]⁺ and its enantiomer, 1.2 g of **2al** and *ent*-**2al** were produced in 89% yield with 93% ee and in 90% yield with 91% ee, respectively. 1.0 g of **2ag** and *ent*-**2ag** were produced in 78% yield, 91% ee and 78% yield, 92% ee, respectively using the same procedure. Dealkylation

of the aryl alkyl ethers **2al** with concentrated boron tribromide led to chiral naphthol **3**[71] in 71% yield 93% ee. Subsequent hydrogenation of aryl benzyl ethers **2ag** and *ent*-**2ag** with the combination of Pd(OH)$_2$ and Pd/C in a 1:1 ratio[72] afforded axial and planar chiral naphthol **3** in 86% yield 92% ee and *ent*-**3** in 87% yield 93% ee, respectively.

To unravel the origin of the high enantioselectivity of the reaction (Fig. 4b), the asymmetric hydroarylations of 1-ethynyl-2-methylnaphthalene derivative **1ao**, 2-benzyne-1-ferrocenylbenzenes **4**, 2-aryne-1-arylbenzenes **5a** and **5b** were also carried out under standard conditions. The **2bo** be obtained in 91% yield with 78% ee and 5:1 *d.r.*, **6** with only the planar chirality was delivered in 99% yield but with low ee (22% ee), in contrast, axial chiral **7a** and **7b** were obtained in 99% yield with 80% ee and 95% ee, respectively and the reactions were complete in less than one minute. Moreover, the linear relationship (see Supplementary Information (SI), Supplementary Fig. S2) between the ees of the Au(TY5)Cl and those of product **2al** and the e.e. of the **2al** did not significantly change during the reaction, which reveal that the enantioselectivity-determining step might involve a single chiral sulfinamide phosphine ligand and one gold species. In light of the structures of the chiral gold catalyst Au[(*Sc,Rs*)-TY5]⁺ and the product **2**, a

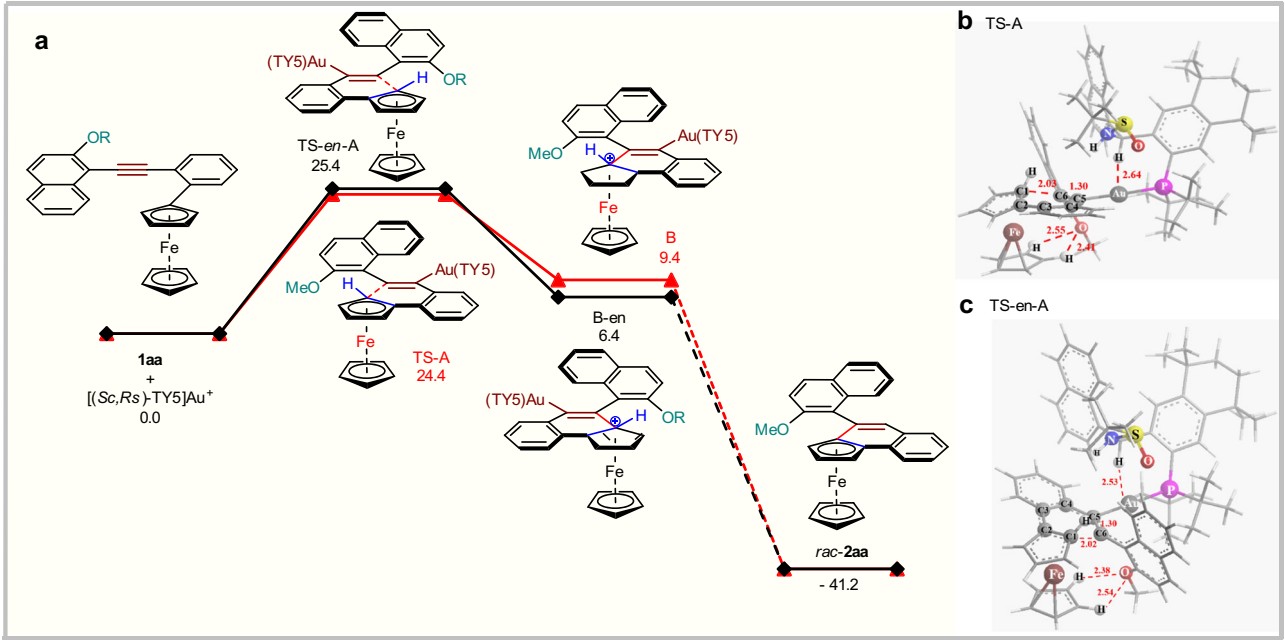

**Fig. 5 Density functional theory (DFT) calculations. a** The Free-energy reaction profiles (kcal mol⁻¹) for the reaction of **1aa**, calculated with SMD Model (dichloromethane) using M062X at 273 K. **b** and **c** The optimized transition states TS-A, TS-en-A for the enantioselectivity-determining step, calculated with SMD Model (Dichloromethane) with M062X method at 273 K.

catalytic chirality-induction model was proposed for the reaction (Fig. 4c). The phosphine of the ligand and the alkyne coordinate to the Au(I) center. The *Re* face of alkyne and the alkoxyl group are shielded by the 1-naphthyl group of the ligand and the ferrocene group attack takes place at the *Si* face to define the (*Sp*)-planar chirality (path b), the alkoxyl group causes a blockage in the rotation of the naphthalene ring to define the (*Sa*)-axial chiralities. Because of these, it defines to form the product (*Sa,Sp*)-**2** with excellent enantioselectivity and diastereoselectivity.

To shed light on the mechanism of the reaction, especially the planar enantioselectivity determining step catalyzed by Au/(*Sc, Rs*)-TY5, density functional theory (DFT) calculations were carried out with Gaussian 09 software package[73–75]. The geometry optimization and frequency calculations were carried out with M062X method and combined basis sets. That is, 6–31G (d) for the reactant **1aa** fragment (except the hydrogen atom of the reaction site on the phenyl ring), the heteroatoms P, S, N, O (connected with S atom) on the ligand (*Sc,Rs*)-TY5 and the carbon atoms linked with the above mentioned heteroatoms, SDD for Au and Fe atoms[76]. Considering the existence of hydrogen bonding, 6–31 + G(d,p) basis set was applied for the H atom of the reaction site on the phenyl ring of **1aa** and the H atoms on the chiral carbon and on the N atom of (*Sc,Rs*)-TY5. 3–21 G basis set was used for all the other atoms. Truhlar and co-workers' SMD solvation model was employed to consider the solvent effect of dichloromethane ($\varepsilon = 8.93$)[77]. The geometry optimizations were performed without symmetry constraints and the nature of the extrema was checked by analytical frequency calculations. The intrinsic reaction coordinate calculation[78,79] was also performed to verify the connectivity of the transition state and the energy minima. (see Supplementary Data for details)

The reaction of **1aa** is selected as the model and we focused on the planar enantioselectivity of the reaction (Fig. 5). Enantio-determining step of the reaction is the 6-*endo*-dig cyclization, that is, aromatic ferrocene attacks the alkyne moiety of **1aa**, which is activated by cationic Au/(*Sc,Rs*)-TY5. The following proto-demetalation gives high enantioselective **2aa**. Barriers (**TS-A** and **TS-en-A**) for the enantioselective step are 24.4 and 25.4 kcal/mol for

the major and minor intermediates B and B-en, which means the reaction can proceed smoothly at reaction temperature. In addition, the difference of the two barriers are 1.0 kcal/mol, that is also in good line with the experimental 85% ee value of **1aa** reaction. Both TSs are late transition states, and the structures are closer to those of B and B-*en* (Fig. 5). For example, in both TSs, new 6-membered rings almost formed, C5-C6 distances are 1.30, very similar to that of normal double bonds. Meanwhile, weak interaction can be found between the tertiary hydrogen atom of TY5 and Au atom, the Au-H distances are 2.64 and 2.53 Å, respectively (Fig. 5). The two hydrogen bonds between O atom of the methoxy group and two hydrogen atoms on ferrocene may contribute to the axial enantioselectivity of the reaction. In TS-A, π-π stacking effect was found between two naphthyl groups from TY5 and **1aa** parts respectively, while there is no such effect in TS-en-A. This may be the cause of high facial enantioselectivity of the reaction.

In summary, we developed an efficient gold(I)/TY-Phos-catalyzed intramolecular hydroarylation of *ortho*-alkynylaryl ferrocenes derivatives, which represents the highly enantioselective and diastereoselective simultaneous construction of axial and planar chiral (*Sa,Sp*)-naphthalene-fused ferrocene derivatives. The axial biaryl compound could be also delivered efficiently under the same reaction conditions. The here identified cationic Au(TY-Phos)⁺ is responsible for the high yield and diastereoselectivity, good to excellent enantioselectivities. The DFT calculations explained the chirality-induction model and accounts for the high enantioselectivity. We believe that this well-designed, easily available gold catalyst Au(TY-Phos)Cl can be applied in other catalytic asymmetric transformations.

## Methods
**Typical procedure for simultaneous construction of axial and planar chirality by Gold/TY-Phos-catalyzed asymmetric hydroarylation.** In a dried Schlenk tube, after the solution of (*Sc,Rs*)-Au(TY5)Cl (5 mol%, 8.2 mg) and NaBArF (6 mol%, 10.6 mg, cas: 79060-88-1, *Energy Chemical*, white powder) in DCM (0.5 mL) was stirred at room temperature for 15 min. Then the above catalyst solution was added to the solution of **1**, **4–5** (0.2 mmol) in DCM (1.5 mL) at 0 °C. The reaction was determined by TLC analysis, after the **1**, **4–5** was consumed completely. Solvent was removed in a rotary evaporator, purified by flash column chromatography on silica

gel (Hexane/DCM = 10:1 to 5:1) to afford the desired product **2**, **6**, **7**. All new compounds were fully characterized (see the Supplementary Information).

## Data availability

The data that support the findings of this study are available within the article, its Supplementary Information files and Supplementary Data files. All data underlying the findings of this work are available from the corresponding author upon reasonable request. The X-ray crystallographic coordinates for structures reported in this study have been deposited at the Cambridge Crystallographic Data Center (CCDC), under deposition numbers 2052064 ((*Sa*,*Sp*)-**2bb**) and 2052077 (((*Sc*,*Rs*)-TY6)AuCl). The data can be obtained free of charge from The Cambridge Crystallographic Data Center via http://www.ccdc.cam.ac.uk/data_request/cif.

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

## Acknowledgements

We gratefully acknowledge the funding support of NSFC (22031004, 21921003), Shanghai Municipal Education Commission (20212308), and the postdoctoral research fund CPSF (2019M661420). We greatly appreciate Ph.D. Yanfei Niu at East China Normal University for her kind help with X-ray single crystal structural analyses, M.S. Lian-Fang Yang at East China Normal University for her kind help with NMR structural analyses.

## Author contributions

J.Z., P.-C.Z., and H.-H.W., conceived the project, analyzed the data and wrote the paper. P.-C.Z., performed the most of experiments. Y.-L.L. and J.H. helped in synthesis of substrates **1**. Z.L. did the DFT calculations. All authors discussed the results and commented on the paper.

## Competing interests

The authors declare no competing interests.
