## [Peer Review File · Nature Communications]

Reviewers' Comments:

Reviewer #1:

Remarks to the Author:

This manuscript describes the simultaneous construction of planar and axial chirality by Au-catalysed hydroarylation of a conveniently designed alkyne, which contains a ferrocene substituent at one of its termini. After ligand and substrate optimization excellent yields, diastereoselectivities and enantioselectivities are obtained. In general, I do believe that the authors did a nice work and would recommend its publication in *Nature Communications* once they tackle the following points:

- Practical utility: In order to get excellent *de* and *ee*, the authors need to include a cyclohexylmethyl (CHM) protecting group at the naphthol unit; this is perfect. However, in order to show further synthetic utility, the authors need to develop a method to deprotect efficiently that naphthol (Maybe, BBr_3 ?). The example shown as example for deprotection in table 3 has a Bn-group, which is easy to deprotect but does not provide excellent *de*'s. If the CHM group cannot be cleanly deprotected, the practical utility of the method will be significantly reduced.
- I find the chirality induction model described in lines 136-141 quite speculative. Some calculations might help here.
- Sometimes the paper is difficult to read. For example lines 36-37: "Inspired by these seminal works, and the simultaneous construction of two different types of chiralities is now flourishing developed"... or line 42: "efficient catalytic asymmetric intramolecular desymmetric cyclisation..."
- The IR spectra of the substrates/products have not been measured.
- Other minor comments. Line 9: "poses" ; Line 127: "The 6a"; Line 202: "(a)", but there is only reference.

Reviewer #2:

Remarks to the Author:

The manuscript describes the enantioselective gold(I)-catalyzed hydroarylation of ortho-alkynylferrocenes. The synthesis of planar-chiral ferrocenes using asymmetric Au(I)-catalysis was already developed (Ref 11), but the reactivity described here is new and interesting on several points. Firstly, the development of catalytic strategies for the installation of two sources of chirality (planar and axial) simultaneously is quite rare. The reaction was able to demonstrate a great versatility (> 40 examples), with different substrates, and always with impressive yields, diastereoselectivities and enantioselectivities. It is worth noting that these good results in catalysis were obtained with "homemade" chiral phosphines.

Thanks to the interesting new reactivity in asymmetric gold catalysis, this article could be accepted for publication in this journal, however with some consideration to the following points:

1. The authors highlighted the role of H-bonding in the catalyst, to explain the good results obtained in this transformation. The authors used the N-Me-TY5 catalyst and showed a decrease in the *ees*. That could be very interesting to also test other substrates such as 1-ethynyl-2-methylnaphthalene or 2-bromo-1-ethynyl-naphthalene derivatives, instead of always 1-ethynyl-naphthalen-2-ol derived substrates. Perhaps this could have an impact on both the yields and the selectivity of the reaction, and it could lead to a better understanding of the limits of this transformation.
2. Concerning the absolute configuration of the product 2bb, both the X-ray figure in Table 2 and the induction model in Table 3c are troublesome. In Table 2, the "Chemdraw" representation of 2bb should be kept like in the SI, page S3. Concerning the induction model, it is very difficult to understand with this scheme that the "path b" is favored. Is it possible to draw both of the possibilities and the corresponding products? Did the authors try to use DFT calculations to evaluate both cycloisomerization pathways? Would it be possible as additional information to give the enantioselectivity of the minor diastereoisomer of at least 2aa?
3. In Table 3a and in the SI, page S41, the authors showed "synthetic applications" of the products 2ag and 2al. From my point of view, this is just a simple deprotection of the OBn or the OCHM substrates. What would be the most useful applications of these chiral molecules? Is there an interest in catalysis or in material sciences? At least one sentence and references could be

added to inform the reader of the interest of accessing these complex molecules.

4. Some references could be added, concerning planar-chiral and helical Ferrocenes:

a) M. Ito, M. Okamura, K. S. Kanyiva and T. Shibata, *Organometallics*, 2019, 38, 4029; b) A. Urbano, A. M. del Hoyo, A. Martínez-Carrión and M. C. Carreño, *Org. Lett.*, 2019, 21, 4623; c) Previous work of the same group : P.-C. Zhang, Y. Wang, D. Qian, W. Li and J. Zhang *Chin. J. Chem.* 2017, 35, 849.

What are the "Ref. 2a and 2b" in the Reference 50?

5. In the SI, some compounds are not pure enough, as can be seen on HPLC Spectra: 2ai (page S21-S22); 2am (page S24); 2an (page S24-S25); 2ao (page S25). Which is this third product in the HPLC spectra, since the diastereoselectivity is >20/1 and the conversion seems total?

6. Carefully check the typos:

line 52: "Carreno" instead of "Carreon"

line 54: "asymmtric"

line 62: "performance in asymmetric" instead of "performance of in asymmetric"

line 64: "85% ee" instead of "82% ee"?

line 73: "entries 8, 11-13" instead of "entries 6, 11-13"

Table 1, entry 1 and line 83 "Carreno"

Table 1, entry 16: add "g"

line 80: you use 10 mol% or 5 mol% of catalyst?

line 117: "hydrogenation" instead of "hydrogenation"

Table 3b: "NaBARF"

In conclusion, the relevance of the work being important, that is why I have a positive opinion for the publication of this paper in this journal.

Reviewer #3:

Remarks to the Author:

The authors disclosed highly enantio- as well as diastereoselective hydroarylations to construct both axial and planar chiralities simultaneously. Au-catalyzed asymmetric hydroarylations were investigated deeply and excellent results were presented. Chiral ligands were carefully screened and TY-phos ligands originated from the author's group proved to be better than other ones. However, I am concerning about the following problems. They should be solved before I can review the manuscript again.

1. The structure of the proposed species shown in Table 3c is rather confusing and could be wrong. It is too crowded to realize. Obviously, path a should be more feasible than path b. Either the chiral ligand or the major diastereomer of compound 2 was assigned a wrong structure by the authors. I suggest the authors to draw a reasonable one according to the crystal structure of compound 2bb (supposing it is correct) as well as the file TS.c3.xml I provided. Please be aware that the cyclohexylmethoxy group plays an important role to control the stereoselectivities. A transition state, in which the cyclohexylmethoxy group lies on the top of the substituted Cp ring, should be highly disfavored.

2. The crystal structure shown in Table 2 is inconsistent with that associated with the CIF file of compound 2bb. Quite possibly it is for the minor diastereomer.

Response to Reviewers: Manuscript NCOMMS-21-03339

Reply to the comments and suggestions of the reviewer 1:

This manuscript describes the simultaneous construction of planar and axial chirality by Au-catalysed hydroarylation of a conveniently designed alkyne, which contains a ferrocene substituent at one of its termini. After ligand and substrate optimization excellent yields, diastereoselectivities and enantioselectivities are obtained.

In general, I do believe that the authors did a nice work and would recommend its publication in nature Communications once they tackle the following points:

- 1) Practical utility: In order to get excellent de and ee, the authors need to include a cyclohexylmethyl (CHM) protecting group at the naphthol unit; this is perfect. However, in order to show further synthetic utility, the authors need to develop a method to deprotect efficiently that naphthol (Maybe, BBr_3 ?).

·Reply: According to your nice suggestion, we have developed the method to deprotect efficiently that naphthol (CHM protecting group) by use of the BBr_3 , seeing the manuscript (Table 3) and SI page 43, and your suggestion really help us to solve the problem.

- 2) I find the chirality induction model described in lines 136-141 quite speculative. Some calculations might help here.

·Reply: Thanks. We took three months to do DFT calculation and to explain the chirality induction model, it is painful but helpful to understand the selectivity. Please find details in the manuscript page 7-9, Line 121-154 and SI page 43-52.

- 3) Sometimes the paper is difficult to read. For example lines 36-37: “Inspired by these seminal works, and the simultaneous construction of two different types of chiralities is now flourishing developed”... or line 42: “efficient catalytic asymmetric intramolecular desymmetric cyclisation...”

·Reply: According to your nice suggestions, we have revised the text and hope it is easy to read now.

- 4) The IR spectra of the substrates/products have not been measured.

·Reply: The IR spectra of the substrates and products have been measured and added to the Supporting Information.

- 5) Other minor comments. Line 9: “poses”; Line 127: “The 6a”; Line 202: “(a)”, but there is only reference.

·Reply: Thank you for pointing this out. We have corrected these mistakes, please find in the manuscript Line 157, Line 158, Line 239, Line 300, Line 310, Line 340, Line 349.

Reply to the comments and suggestions of the reviewer 2:

The manuscript describes the enantioselective gold(I)-catalyzed hydroarylation of ortho-alkynylferrocenes. The synthesis of planar-chiral ferrocenes using asymmetric Au(I)-catalysis was already developed (Ref 11), but the reactivity described here is new and interesting on several points. Firstly, the development of catalytic strategies for the installation of two sources of chirality (planar and axial) simultaneously is quite rare. The reaction was able to demonstrate a great versatility (> 40 examples), with different substrates, and always with impressive yields, diastereoselectivities and enantioselectivities. It is worth noting that these good results in catalysis were obtained with “homemade” chiral phosphines.

In conclusion, the relevance of the work being important, that is why I have a positive opinion for the publication of this paper in this journal. Thanks to the interesting new reactivity in asymmetric gold catalysis, this article could be accepted for publication in this journal, however with some consideration to the following points:

- 1) The authors highlighted the role of H-bonding in the catalyst, to explain the good results obtained in this transformation. The authors used the N-Me-TY5 catalyst and showed a decrease in the ees. That could be very interesting to also test other substrates such as 1-ethynyl-2-methylnaphthalene or 2-bromo-1-ethynyl-naphthalene derivatives, instead of always 1-ethynyl-naphthalen-2-ol derived substrates. Perhaps this could have an impact on both the yields and the selectivity of the reaction, and it could lead to a better understanding of the limits of this transformation.

*·Reply: Thanks for your kind suggestions. We have synthesized the substrate 1-ethynyl-2-methylnaphthalene derivatives **1bo** (Table 3). **1bo** were carried out under standard conditions, could deliver the desired products **2bo** in 91% yields with 78% ee and 5:1 dr. It indicates that the steric hindrance of R group, rather than hydrogen bonding, plays an important role to control the stereoselectivities, please find the detail in Table 1, in the manuscript.*

- 2) Concerning the absolute configuration of the product 2bb, both the X-ray figure in Table 2 and the induction model in Table 3c are troublesome. In Table 2, the “Chemdraw” representation of 2bb should be kept like in the SI, page S3. Concerning the induction model, it is very difficult to understand with this scheme that the “path b” is favored. Is it possible to draw both of the possibilities and the corresponding products? Did the authors try to use DFT calculations to evaluate both cycloisomerization pathways? Would it be possible as additional information to give the enantioselectivity of the minor diastereoisomer of at least 2aa?

*·Reply: We are very sorry for providing a misleading crystal structure shown in Table 2. We redraw a distinct one according to the crystal structure of compound 2bb (Table 2). We also redraw both of the possibilities and the corresponding products (Table 3). We have carried out DFT calculation to explain the chirality induction model. The two hydrogen bonds between O atom of the methoxy group and two hydrogen atoms on the bottom of the substituted Cp ring may contribute to the axial enantio-selectivity of the reaction. And π - π stacking effect in **TS-A** was*

found between two naphthyl groups from TY5 and 1aa parts respectively, while there is no such effect in TS-en-A. This may be the cause of high planar enantio-selectivity of the reaction. Obviously, path a should be more feasible than path b.

- 3) In Table 3a and in the SI, page S41, the authors showed “synthetic applications” of the products 2ag and 2al. From my point of view, this is just a simple deprotection of the OBn or the OCHM substrates. What would be the most useful applications of these chiral molecules? Is there an interest in catalysis or in material sciences? At least one sentence and references could be added to inform the reader of the interest of accessing these complex molecules.

·Reply: Thanks for your nice suggestions. We think that the most useful applications of these chiral molecules are as chiral catalyst. The potential applications of these chiral molecules are also interest in asymmetric catalysis and material sciences. In order to indicate the potential applications of these chiral molecules, the sentence and references have been added to inform the reader of the interest of accessing these complex molecules (Reference 73).

- 4) Some references could be added, concerning planar-chiral and helical Ferrocenes:
a) M. Ito, M. Okamura, K. S. Kanyiva and T. Shibata, *Organometallics*, 2019, 38, 4029; b) A. Urbano, A. M. del Hoyo, A. Martínez-Carrión and M. C. Carreño, *Org. Lett.*, 2019, 21, 4623; c) Previous work of the same group: P.-C. Zhang, Y. Wang, D. Qian, W. Li and J. Zhang *Chin. J. Chem.* 2017, 35, 849.

What are the “Ref. 2a and 2b” in the Reference 50?

·Reply: We are very sorry that we missed these relevant literatures. These references have been added, and the Reference 50 have been corrected, seeing the manuscript References 13, 40, 51, 52.

- 5) In the SI, some compounds are not pure enough, as can be seen on HPLC Spectra: 2ai (page S21-S22); 2am (page S24); 2an (page S24-S25); 2ao (page S25). Which is this third product in the HPLC spectra, since the diastereoselectivity is >20/1 and the conversion seems total?

·Reply: To evaluate this question, we did some control experiments. It was obvious that some compounds (2) was unstable in isopropanol, which may be the cause of the third product in the HPLC spectra. The new HPLC spectra (2ai, 2am, 2an, 2ao) were provided in SI, by quick HPLC analysis.

- 6) Carefully check the typos:
line 52: “Carreno” instead of “Carreon”
line 54: “asymmtric”
line 62: “performance in asymmetric” instead of “performance of in asymmetric”
line 64: “85% ee” instead of “82% ee”?

line 73: "entries 8, 11-13" instead of "entries 6, 11-13"
Table 1, entry 1 and line 83 "Carreno"
Table 1, entry 16: add "g"
line 80: you use 10 mol% or 5 mol% of catalyst?
line 117: "hydrogenation" instead of "hydrogenation"
Table 3b: "NaBARF"

·Reply: Thank you for pointing this out. We have carefully checked the typos and corrected the above errors.

Reply to the comments and suggestions of the reviewer 3:

The authors disclosed highly enantio- as well as diastereoselective hydroarylations to construct both axial and planar chiralities simultaneously. Au-catalyzed asymmetric hydroarylations were investigated deeply and excellent results were presented. Chiral ligands were carefully screened and TY-phos ligands originated from the author's group proved to be better than other ones. However, I am concerning about the following problems. They should be solved before I can review the manuscript again.

- 1) The structure of the proposed species shown in Table 3c is rather confusing and could be wrong. It is too crowded to realize. Obviously, path a should be more feasible than path b. Either the chiral ligand or the major diastereomer of compound 2 was assigned a wrong structure by the authors. I suggest the authors to draw a reasonable one according to the crystal structure of compound 2bb (supposing it is correct) as well as the file TS.c3xml I provided. Please be aware that the cyclohexylmethoxy group plays an important role to control the stereoselectivities. A transition state, in which the cyclohexylmethoxy group lies on the top of the substituted Cp ring, should be highly disfavored.

*·Reply: Many Thanks. We are very sorry for providing a misleading crystal structure shown in Table 2. We redraw a distinct one according to the crystal structure of compound **2bb** as well as the file TS.c3xml. (See Table 2). As you mentioned, I agree with you for the most part, we redraw both of the possibilities and the corresponding products in Table 3c. We have carried out DFT calculation to explain the chirality induction model. The two hydrogen bonds between O atom of the methoxy group and two hydrogen atoms on the bottom of the substituted Cp ring may contribute to the axial enantio-selectivity of the reaction. And π - π stacking effect in **TS-A** was found between two naphthyl groups from **TY5** and **Iaa** parts respectively, while there is no such effect in **TS-en-A**. This may be the cause of high planar enantio-selectivity of the reaction. Obviously, path a should be more feasible than path b.*

- 2) The crystal structure shown in Table 2 is inconsistent with that associated with the CIF file of compound 2bb. Quite possibly it is for the minor diastereomer.

*·Reply: We are very sorry for providing a misleading crystal structure shown in Table 2. We redraw a distinct one according to the crystal structure of compound **2bb***

as well as the file TS.c3xml. (See Table 2)

Reviewers' Comments:

Reviewer #1:

Remarks to the Author:

I think that the manuscript has improved significantly after the first revision and that it can now be published (I am not an expert in calculations, so I cannot evaluate this part, but I appreciate that the authors now propose a more detailed rationale for their results). However, before publishing, I recommend that the authors review the text and diagrams before and eliminate the grammatical errors and other inaccuracies that are present. For example, and without being exhaustive:

-line 15: "calculations"

-line 23: "the construction of axially chiral construction"

-lines 37-38: "...flourishing developed..."

-lines 66-67: "...a more sigma-donating ligands..."

-lines 85-86: "... no counterion or only additives all haven't catalytic activity"

-line 162: "2-arynyl"

-line 164: "The 2ao"

-Figure 3 "BB3"

Reviewer #2:

Remarks to the Author:

With the additional experiments and DFT calculations this study is more comprehensible. Concerning the induction model, the Table 3c is more clear like that. I just have a « graphical » comment, which is also related to the accuracy of the intermediates. In Figure 2 [intermediates TS-en-A, TS-A, B and B-en], and also in Table 3c [intermediates (Sa,Sp)-B and (Sa, Rp)-B-en] the Cp ferrocene backbone should be redrawn. Indeed, if you keep the « circle » in the Cp ring, you have a valence problem. To avoid that, you should instead draw two double bonds, like in Scheme 5 of this publication [JOC 81, 6266 (2016), reference 12].

Typos : « BBr3 » instead of (BB3) in Table 3a.

Reviewer #3:

Remarks to the Author:

The authors carefully revised and improved the manuscript and my concerns about the transition state as well as the absolute configuration of the major diastereomer have been resolved. I recommend publication of the paper in the present form.

Reply to the comments and suggestions of the reviewer 1:

I think that the manuscript has improved significantly after the first revision and that it can now be published (I am not an expert in calculations, so I cannot evaluate this part, but I appreciate that the authors now propose a more detailed rationale for their results). However, before publishing, I recommend that the authors review the text and diagrams before and eliminate the grammatical errors and other inaccuracies that are present. For example, and without being exhaustive:

-line 15: “caculations”

-line 23: “the construction of axially chiral construction”

-lines 37-38: “...flourishing developed...”

-lines 66-67: “...a more sigma-donating ligands...”

-lines 85-86: “... no counterion or only additives all haven't catalytic activity”

-line 162: “2-arynyl”

-line 164: “The 2ao”

-Figure 3 “BB₃”

·Reply: Thank you for pointing this out. We have carefully checked the typos and corrected the above errors.

Reply to the comments and suggestions of the reviewer 2:

With the additional experiments and DFT calculations this study is more comprehensible. Concerning the induction model, the Table 3c is more clear like that. I just have a « graphical » comment, which is also related to the accuracy of the intermediates. In Figure 2 [intermediates TS-en-A, TS-A, B and B-en], and also in Table 3c [intermediates (Sa,Sp)-B and (Sa, Rp)-B-en] the Cp ferrocene backbone should be redrawn. Indeed, if you keep the « circle » in the Cp ring, you have a valence problem. To avoid that, you should instead draw two double bonds, like in Scheme 5 of this publication [JOC 81, 6266 (2016), reference 12].

·Reply: According to your nice suggestions, the Cp ferrocene backbone have been redrawn.

Typos: « BBr₃ » instead of (BB₃) in Table 3a.

·Reply: Thank you for pointing this out. We have carefully checked the typos and corrected the above errors.

Reply to the comments and suggestions of the reviewer 3:

The authors carefully revised and improved the manuscript and my concerns about the transition state as well as the absolute configuration of the major diastereomer have been resolved. I recommend publication of the paper in the present form.

·Reply: We appreciate the time and effort that you dedicated to providing feedback on our manuscript and are grateful for the insightful comments on and valuable improvements to our paper.